# Effects of Feed Supplementation on Nesfatin-1, Insulin, Glucagon, Leptin, T3, Cortisol, and BCS in Milking Ewes Grazing on Semi-Natural Pastures

**DOI:** 10.3390/ani11030682

**Published:** 2021-03-04

**Authors:** Olimpia Barbato, Elena De Felice, Luca Todini, Laura Menchetti, Alessandro Malfatti, Paola Scocco

**Affiliations:** 1Department of Veterinary Medicine, University of Perugia, via San Costanzo 4, 06126 Perugia, Italy; olimpia.barbato@unipg.it; 2School of Biosciences and Veterinary Medicine, University of Camerino, via Gentile III da Varano, 62032 Camerino, Italy; luca.todini@unicam.it (L.T.); paola.scocco@unicam.it (P.S.); 3Department of Agricultural and Food Science, University of Bologna, viale G. Fanin 44, 40137 Bologna, Italy; laura.menchetti7@gmail.com

**Keywords:** feed supplementation, *Ovis aries*, metabolic hormones, BCS, drought stress, Italian Central Apennine

## Abstract

**Simple Summary:**

In Central Italy Apennine sheep represent the most bred species, which is still bred today in a semi-extensive manner, exploiting the natural pastures. However, the increase in summer aridity induces a decrease in the pastoral value of the grasslands, resulting in poor animal performance and production. Only research-derived innovation can support farmers’ economy in order to maintain the pastoral activities. The objective of this study was to evaluate the effects of cereals supplementation on body condition score and metabolic hormones profile in milking ewes grazing on semi-natural pastures on mid mountain rangelands. Our findings showed that feed supplementation preserves grazing ewes from the usual lowering of the body state associated to the lactation period and sustains the metabolic status of animals. Taken together, our results could represent a helpful instrument in the farm management practices.

**Abstract:**

This study aimed to investigate the effects of feed supplementation on body condition score (BCS) and different metabolic hormones concentration in lactating sheep reared in Italian Central Apennine pastures during the grazing summer period. In this study, 24 Comisana x Appenninica pluriparous ewes from June until August were divided into two homogenous groups: the control group (UNS) was free to graze, while the other group (SUP), in addition to grazing, was supplemented with 600 g/day/head of cereals. At the start of the supplementation and at an interval of 9–10 days until the end of experimentation, BCS evaluation and blood withdrawal to assay nesfatin-1, insulin, glucagon, leptin, triiodothyronine and cortisol levels were performed. Univariable analysis showed no remarkable differences between the groups, while multivariable analysis suggested that the UNS group was characterized by a lower BCS and greater nesfatin-1 than the SUP group. These findings can be considered in relation to milk production, which shows a clear better persistence in the SUP group. Our results indicate that nutritional supplementation has protected ewes from the usual lowering of the body state linked to lactation and provides a good maintenance of milk production, determining also a better overall body and metabolic state of the animals, which is important at the beginning of the sexual season.

## 1. Introduction

It is well known how the feeding strategies and the state of nutrition are fundamental to improve productions in the livestock species. In the sub-Mediterranean areas, drought stress, due to the increasing summer aridity, is progressively decreasing the pastoral value of natural and semi-natural pastures used as a trophic resource for ovine semi-extensive breeding, thus negatively reflecting on animal status [1,2,3]. This fact could be taken into account in the management of flocks reared in Apennine pastures during the spring–early autumn period, in which global warming is causing an advance of both the pasture flowering peak and the pasture dryness [4,5].

A better nutrition status can improve the productive and reproductive efficiency through numerous factors, including the circulation of hormones as well as nutrient-sensitive metabolites [6]. It has been stamped that endocrine and neuroendocrine events play a fundamental role in regulating food intake and energy homeostasis. As a consequence of the absorption of nutrients and/or changed metabolism, the blood concentration of metabolic hormones can be changed. Furthermore, the process of digestion can induce changes in different hormones due to mechanical or chemical stimulation of receptors in the digestive tract.

Unlike other metabolic hormones, nesfatin (NES1) was identified only relatively recently, and there is still poor knowledge about its role in metabolism in ruminant animals. It was recognized in 2006 as a potent anorexigenic peptide involved in the regulation of homeostatic feeding [7]. Subsequently, its widespread central and peripheral distribution gave rise to additional effects. In fact, NES1 exerts pleiotropic actions at the levels of digestive systems [8], energy and glucose homeostasis [9]. Moreover, it is involved in stress response [10], sleep [11] and reproduction [12,13] in both humans and different animal species.

Pancreatic insulin (INS) is a main endocrine signal in the control of nutrient partitioning and the metabolism of carbohydrates, proteins and lipids. It has a role in homeorhetic processes, which allow the animal to adapt the nutrient partitioning to changes of physiological states and nutrient requirements [14]. In lactating dairy goats, plasma INS levels were positively correlated with energy intake and negatively with dry matter intake [15]. In lactating dairy cows, insulin positively regulates plasma leptin [16].

Glucagon, the other major pancreatic hormone, plays a key role in glucose homeostasis due to its stimulating effect on hepatic glucose output in response to low blood glucose levels [17]. Glucagon has also been suggested to stimulate lipolysis in adipose tissue [18] and to provide a satiety signal [19].

Leptin (LEP), the hormone mainly produced by the adipose tissue, regulates food intake and energy expenditure [20]. Leptin expression and secretion are correlated with body condition, physiological status (puberty, pregnancy, lactation), age, and level of nutrition [21,22]. Furthermore, the presence of leptin receptors at the mammary level [23] suggests a potential role in stimulating mammary development and functions.

The appropriate thyroid gland function is considered crucial to sustain the productive performance in domestic animals (growth, milk, or hair fiber production). Variations of thyroid hormones bioactivity allow animals to adapt their metabolic balance to different environmental conditions, changes in nutrient requirements and availability, and to homeorhetic changes during different physiological stages [24].

The main functions of adrenal cortisol (COR) are to induce protein and fat catabolism, supporting gluconeogenesis. COR makes available body reserves inducing hyperglycemia and providing energy during the stress response. In the current context of climate changes, heat stress can involve animals also in Mediterranean areas, during the summer hotter months, when dairy small ruminants are lactating and nutrient supplies are not always optimal. The effect of heat and nutritional stress on COR secretion is well documented in sheep [25].

The body condition score (BCS) is a more sensitive indicator of the nutritional status of the animals than body weight, which carries also the contribution of the gastrointestinal contents and can lead to an incorrect estimation (over- or under-) of the animal status and welfare [26].

Therefore, our study was undertaken to investigate the effects of feed supplementation during the second phase of lactation on body condition and different metabolic hormones blood concentration in lactating sheep reared in the Italian Central Apennine pastures during the summer late spring–early autumn grazing period.

In addition, to our knowledge, until now, there are no studies relating to nutritional status and nesfatin plasma concentration in sheep. Findings resulting from our research could have some future benefits, such as improving the evaluation of the opportunity to design the most appropriate diets or if a supplementation strategy can be considered an appropriate investment by the breeders.

## 2. Materials and Methods

### 2.1. Location, Animals, and Diets

The trial was carried out on 24 Comisana × Appenninica pluriparous ewes. Lactating ewes were free to graze from the beginning of June to the end of August on a semi-natural pasture located in Central Appennine (Monte Cavallo, altitude 1280 m a.s.l.; longitude 12°58′53″ Est; latitude 42°59′55″ Nord, Marche region, Italy). The animals at the moment of the offspring separation (near 40 d post-partum) were divided in two homogeneous groups as far as body weight (BW) and body condition score (BCS), milk yield, parity, and days of lactation (see Appendix A). One group (SUP, n = 12, mean BW 49.4 ± 6.1, mean BCS 2.31 ± 0.3) was supplemented with 600 g/day/animal of corn and barley (1:1), while the other group (control unsupplemented group, UNS, n = 12, mean BW 50.5 ± 5.1, mean BCS 2.27 ± 0.3) was fed only with the pasture (Appendix A). The feed supplementation was administered during the morning milking when each animal was in its milking post. The feed supplementation was chosen taking into account the management habits of the farmer and in order to avoid too expensive actions.

The food intake of grazing animals was evaluated by the pasture phytomass removal estimation [27].

Milk production was monitored at three different times: at the beginning (7 July 2016, near 75 days after parturitions, morning milking mean yield 297 ± 63 mL by UNS and 306 ± 64 mL by SUP ewes), in the middle (21 July 2016, 134 ± 47 mL by UNS and 240 ± 52 mL by SUP ewes), and in the end (9 August 2016, 125 ± 51 mL by UNS and 192 ± 52 mL by SUP ewes) of the period of nutritional supplementation as previously described [28].

### 2.2. BCS, Blood Collection, and Assays

Hormone levels and BCS were evaluated before the supplementation on 7 July (T0), and they were then measured every 9–10 days until 22 August (T1–T5) (Appendix A). The last date corresponded to the pre-mating period, when males were introduced in the flock and all females were dried. BCS was evaluated on the basis of a specific sheep method previously described [2,3,4,5,6,7,8,9,10,11,12,13,14,15,16,17,18,19,20,21,22,23,24,25,26,27,28,29]. Briefly, BCS is a comprehensive assessment of animal’s body status based on muscle/fat relative proportions, and it is considered a useful management tool in determining the welfare of domestic animals. The steps in ovine BCS evaluation are palpation of body structures (processes of the thoracic and lumbar vertebrae), the state of the dorso-medial area, and the general status of animals. Each step, which was done by 3–5 trained valuators, receives a score (0–5) [2], and the mean of the scores for the four parameters constitutes the BCS value. Blood samples were withdrawn by jugular venipuncture in evacuated tubes containing K3-EDTA as anticoagulant (Sarsted, Numbrecht, Germany). Tubes were immediately centrifuged (2500× *g* for 15 min), and the plasma aliquots were stored at −20 °C until assayed. The samplings were carried out in all the dates at 7:00–8:00 a.m.

All hormone concentrations were determined as the average of duplicate determinants. To minimize the systematic error owing to inter-assay variability, all samples of each animal were analyzed for each hormone within the same assay session, in which an equal number of animals belonging to the two groups was present.

Enzyme immunoassays (EIA) were performed using the automated processor Brio 2 reader (Seac, Firenze, Italy). NES1 was assayed on T0, T2, T4 and T5, due to the lack of plasma amount on the other dates. NES1 level was measured using EIA kits (E-E-H2373, Biotechnology Inc., Human NES1, Wuhan, China). The intra-and inter-assay coefficients of variation (CV) were 9.4% and 11.6%, respectively. Sensitivity (DL) indicated by the manufacturer is 9.38 pg/mL. Plasma INS was determined using the Sheep Insulin ELISA kit (EIA-4739, DRG, Marburg, Germany) and the intra- and inter-assay CVs were 4.3% and 9.7%, respectively. Sensitivity (DL) indicated by the manufacturer is 0.49 µLU/mL. Radioimmunoassay was used to measure plasma glucagon (Glucagon RIA Kit, RB310, Euro Diagnostica AB, Lundavagen, Malmo, Sweden). The intra- and inter-assay coefficients of variation for control samples were 7.6% and 5.4%, respectively. Sensitivity (DL) indicated by the manufacturer is 3 pmol/L. Plasma LEP concentrations were measured by the multi-species Leptin RIA kit (XL-85K, EMD Millipore Co., Billerica, MA, USA) and the intra- and inter-assay coefficients of variation were 3.4 and 8.7%, respectively. Sensitivity (DL) indicated by the manufacturer is 0.801 ng/mL Human Equivalent (HE). Total concentrations of 3-3′-5-triiodothyronine (TH) in plasma were assayed using a radioimmunoassay kit (Immunotech, Prague, Czech Republic, IM 3287). Intra- and inter-assay coefficients of variation (CVs) were 6.1% and 7.8%, respectively. Sensitivity (Detection Limit DL) indicated by the manufacturer is 0.26 nmol/L. Plasma cortisol concentrations were determined using commercial RIA kits (Immunotech, Prague, Czech Republic, IM 1841) and the intra- and inter-assay CVs were 7.3% and 9.1%, respectively. Sensitivity (DL) indicated by the manufacturer is 5 nM/L.

### 2.3. Statistical Analysis

Diagnostic graphics and Shapiro–Wilk were used for testing assumptions and outliers. Since non-normality of the data was detected, insulin was log transformed, while NES1 was categorized into two levels using the median [30], as its distribution did not improve after transformation.

First, the effect of time and supplementation were analyzed for each hormone (except for NES1) and BCS (treated as a continuous variable) by using univariable approaches and Linear Mixed Models (LMMs) procedures. Animals and days were included as random and repeated factors, respectively. The LMMs evaluated the main effects of Group (2 levels: UNS and SUP), Time (6 levels: T0–T5), and the interaction between Group and Time. Sidak adjustment was used for carrying out multiple comparisons. Results were expressed as estimated marginal means ± standard error (SE) while raw data were presented in figures as means ± SE.

After categorization, NES1 was analysed by a Generalized Linear Model (GLM) using binomial as the probability distribution and logit as the link function. The effects of group (2 levels: UNS and SUP), sampling time (6 levels: T0–T5), and their interaction were evaluated. Data were presented as percentages, medians (Mdn), and interquartile ranges (IQR).

Finally, a multivariable approach was used by Discriminant Analysis (DA) to find the combinations of variables (BCS and hormones) that distinguish sheep that received supplementation (SUP) from the control group (UNS). The relative importance of each variable on this discrimination was expressed by the Wilks’ lambda (the smaller the Wilks’ lambda, the more important the variable to the Df) and by the discriminant loadings (correlations between each independent variable and the discriminant scores associated with the Df) [31,32]. Mahalanobis distance was used to identify the presence of multivariate outliers. The performance of the DA was evaluated by leave-one-out cross-validation, calculating the probability for each sample to be accurately classified in the correct group. The centroids (mean discriminant scores of the groups) were used to establish the cutting point for classifying samples during the cross-validation.

Statistical analyses were performed with SPSS Statistics version 25 (IBM, SPSS Inc., Chicago, IL, USA). Statistical significance occurred when *p* < 0.05.

## 3. Results

### 3.1. Univariable Approach

Only the time effect was found for BCS, which was significantly reduced at T4 compared to T0 (from 1.7 ± 0.1 at T0 to 1.5 ± 0.1 at T4; *p* < 0.001) (Figure 1) and then returned to basal values T5 (1.8 ± 0.1).

Due to the high variability of the NES1 concentrations (mean ± SE = 121.96 ± 29.09 pg/mL; Mdn (IQR) = 15.86 pg/mL (10.00–65.68 pg/mL); range = 10.00–1000.00 pg/mL), we decided to categorize these data according to their median value. Thus, two categories were created for NES1: “low level” if NES1 < 15.86 pg/mL and “high level” if NES1 ≥ 15.86 pg/mL. No change over time was found for this binary variable (*p* = 0.949), while a significant effect of group was found (*p* = 0.033). Indeed, a higher percentage of NES1 samples from the UNS group were included in the high-level category compared to the SUP group (60.2 ± 7.7% and 36.1 ± 7.7% of samples included in the high-level for UNS and SUP groups, respectively) (Figure 2). Mean values, standard error, and median with an interquartile range for NES1 concentrations in UNS and SUP groups are shown in Appendix A.

The insulin log values increased during T2–T4 times compared to T0 (*p* < 0.001) (Figure 3) in both groups (*p* = 0.795).

Glucagon levels were not affected by either time (*p* = 0.331) or group (*p* = 0.229) (Figure 4).

Overall, leptin levels increased from a marginal mean of 5.1 ± 0.4 ng/mL at T0 to 5.8 ± 0.4 ng/mL at T5 (*p* = 0.016). A significant group x time effect was also found (*p* = 0.028). Indeed, pairwise comparisons revealed that there were no significant changes over time in the UNS group (*p* = 0.252), while the SUP group showed a progressive increase from T2 to the last time point (*p* = 0.002) (Figure 5).

TH concentrations reduced after 10 days (from 3.2 ± 0.1 nmol/L at T0 to 2.8 ± 0.1 at T1; *p* < 0.001) but subsequently stabilized without differences between groups (*p* = 0.491) (Figure 6).

Nutrition treatment affected the trend of cortisol: changes over time were not significant in the UNS group (*p* = 0.111), while the SUP group showed higher values than UNS at T0 (*p* = 0.033), followed by a significant reduction compared to T0 from T2 until T4 (*p* < 0.01) (Figure 7).

### 3.2. Multivariable Approach

Seven variables were included in the DA describing the body condition and hormonal profile of the ewes. One item was eliminated because it was identified as a multivariate outlier by the Mahalanobis distance. The variables that most discriminated the two groups were BCS and NES1 (*p*-value < 0.05) (Table 1). The discriminant loadings (negative for BCS and positive for NES1) (Table 1) and centroids (0.666 and −0.610 for UNS and SUP, respectively) showed that lower BCS and higher NES1 characterized the UNS group compared to the SUP one. Overall, the DA showed a moderate discriminating ability: the model Wilks’ Lambda was significant at *p* < 0.05 level (*p* = 0.045) and the extracted discriminant function correctly classified 63.0% of the samples (cross-validation procedure).

## 4. Discussion

The two ewes groups observed in the present study show very similar changes of the mean BCS value while, considering the univariable model analysis, no significant differences in the BCS changes and hormonal profiles between the two groups were noted.

These findings can be considered in relation to the milk production of the groups, which shows a clear better persistence in the SUP group, as reported in another paper by the same research project [28]. Milk yield was considered as relatively equal on day T0, while it was 78% vs. 45% at day 14 and 63% vs. 42% at day 33, in SUP and UNS, respectively. This observation can drive the hypothesis that the supplementation has protected the ewes from the usual lowering of the body state linked to the lactation, because it could be expected, on the contrary, that a higher and persistent milk production induces a more intense body reserve consumption. Indeed, the different hormones change with very scarce differences between the two groups and without any significant relationships with the body state (except NES1).

The multivariable analysis offers a more overall view of the changes in the considered parameters, weighing up the different effects and influences. In our analysis, the relationship between BCS and NES1 indicates the high probability that a better BCS level, as found in the SUP group, is related to a lower NES1 hematic concentration. This finding confirms that the nutritional supplementation had an effect on the body status, balancing the higher milk production in respect to the UNS group, and that these ewes, probably as a consequence of the nutritional supplementation, maintained a hunger motivation higher than the UNS: the lower blood Nesfatin-1 concentrations drive to suppose a higher hunger sense.

Regarding the changes of the other hormone concentrations, which can be of some interest in sheep grazing in our conditions, we can observe that concentrate supplementation significantly sustained milk production [28], which is often without affecting systemic concentrations of the hormones investigated. Therefore, in such conditions, the larger availability of nutrients in SUP ewes seems addressed toward milk synthesis by adaptations mostly driven at the peripheral (local) level [27].

Insulin participates in sustaining a general anabolic state during pregnancy, while the decrease in INS secretion and/or tissue responsiveness at the beginning of lactation allows the catabolic processes to dominate, and most of the nutrients are shifted toward the mammary gland for milk synthesis. The variations of INS concentrations observed over time in our study well agree with previous findings: an increase in plasma INS concentration has been widely described as lactation progresses, with decreasing milk yield and increasing energy balance [33,34]. The lowest concentrations of INS were found in ewes in early lactation (20 days); thereafter, plasma insulin significantly increased with the advancing stages of lactation (40 and 60 days) [35].

In our trial, glucagon changes were very scarce, and no significant difference was detected between groups. We can speculate that the slight increases in SUP opposite to the slight decrease in UNS can be consistent with a tendency to increase the available energy by its effects on hepatic glucose output and, possibly, lipolysis by adipose tissue.

In the SUP group, LEP concentrations progressively increase after two weeks of supplementation toward the end of lactation (T4) and dry off (T5). Similar to our results of the SUP group, in the ewe, leptinemia is reported to be very low during the first 3 weeks of lactation [36,37]; then, it gradually increases until about 3 months after lambing [38]. The lack of LEP increase with advancing lactation in UNS ewes likely mirrors the lack of improvements of the energy balance in these animals.

The plasma TH concentrations in both groups of ewes significantly lowered at day 85 of lactation, accordingly with previous reports. Thyroid hormone concentrations tended to decrease from 36 h to 21 days post-partum and thereafter constantly rose until day 51 post-partum [39], and slightly higher concentrations of TH were observed in the blood of ewes at the 40th day of lactation than at the 20th day [40]. After the start of our trial, animals were also exposed to rising summer ambient temperatures, which exert a well-known depressive effect on thyroid gland activity and TH action [41] and are inversely correlated with plasma thyroid hormone levels [25,26,27,28,29,30,31,32,33,34,35,36,37,38,39,40,41,42,43]. In the present study, high ambient temperatures may have prevented the physiological recovery of thyroid hormone concentrations, which are often observed toward the end of lactation in small ruminants, in parallel with the reduction of milk production and the increase of energy balance. On the other hand, in early lactation, low TH, leptin and insulin levels can indicate an energy deficiency of ewes, even despite the concentrate supplementation. Blood metabolic hormones and BCS are very useful tools for the assessment of ewes’ nutritional status in very demanding physiological states, such as late pregnancy and lactation [40,41,42,43,44].

COR changes over time were not significant in the UNS group, while the SUP one showed higher values than the UNS group at T0, which was followed by a significant reduction until T4. The higher COR values found in the SUP group at the first sampling are likely due to the very large physiological variations among individuals when hormone concentrations increase facing different situations. For example, the blood-sampling procedure itself in unaccustomed animals could have played a role. Relatively low COR levels during lactation are reported in goats [45]. Regarding the COR peak observed in SUP ewes at the last sampling, different papers report similar effects by heat stress in small ruminants [46,47]. The serum COR levels were the highest during summer [43,44,45,46,47,48,49] and positively correlated with ambient temperature, showing reverse trends and negative correlations with TH [42,43,44,45,46,47,48,49,50]. Moreover, during thermal stress, feed-restricted ewes [51] and rams [25] showed less increase in cortisol blood concentrations: therefore, if the nutritional deficit is added to thermal stress, the animals can adjust their response, avoiding the potential adverse effects of extreme cortisol action, as it could be the case of the UNS group of the present study.

The trial planning foresaw performing further controls, especially aimed to know how the ewes of both groups cope with the successive reproductive season, in the hypothesis that the SUP group could face this demanding phase of the productive cycle performing better than the UNS one. Unfortunately, just at the start of the sexual season, when the males were to be joined in the female flock, in Central Italy a powerful earthquake took place (two strong events at the end of August and at the end of October). The farm where our ewes were kept suffered a lot of damage, as well as the houses of the farmers, so it was not possible to carry on the intended further controls, and the overall situation after the earthquake could have direct and indirect effects influencing the results. Despite this accident, fortunately, we had the possibility to have some indication from research performed on other ewes of the same flock, which showed that the feed supplementation had a positive effect on resistin production in the sheep uterine glands [52] and in the apelinergic system expression in both mammary glands and female reproductive apparatus [27,28,29,30,31,32,33,34,35,36,37,38,39,40,41,42,43,44,45,46,47,48,49,50,51,52,53,54].

## 5. Conclusions

Our results, notwithstanding the lack of the possibility to perform all the forecasted controls, indicate that the nutritional supplementation of lactating ewes reared in semi-extensive conditions on mid mountain rangelands can determine an overall body and metabolic state better than expected in milking animals approaching reproductive activity. Our findings were especially revealed by the combined observations of (a) the ability by the SUP ewes to maintain a BCS value similar to the UNS one, despite higher milk production, and (b) the lower blood levels of NES1 that suggest a better penchant to restore their body reserve in view to face the reproductive engagement. On the contrary, the UNS ewes, due to a higher level of the anorexigenic peptide NES1, could have more difficulties to recover a good nutritional status in view of the sexual activity.

The fact that the circulating levels of some of the studied hormones did not show differences between unsupplemented and supplemented ewes do not indicate necessarily that the supplementation could not had favorable localized effects at the peripheral level. Indeed, recent studies carried out on the same animal groups suggested that the feed supplementation had a positive effect on some adipokine expression in different organ tissues.

## Figures and Tables

**Figure 1 animals-11-00682-f001:**
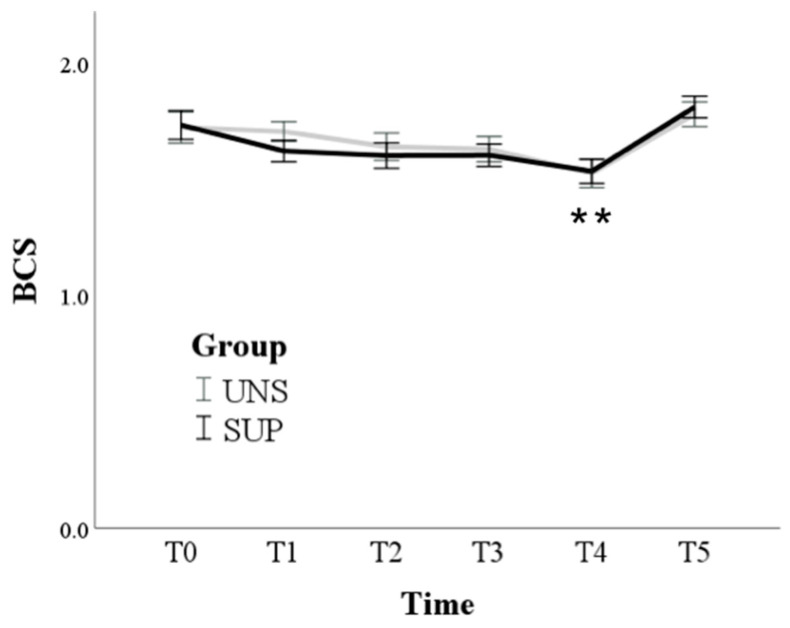
Means and standard errors of body condition score (BCS) in control (UNS) and supplemented (SUP) lactating sheep monitored before the administration of supplementation (T0), and then every 10 days until dry off (T1–T5). ** *p* < 0.01 T4 vs. T0 in SUP and UNS.

**Figure 2 animals-11-00682-f002:**
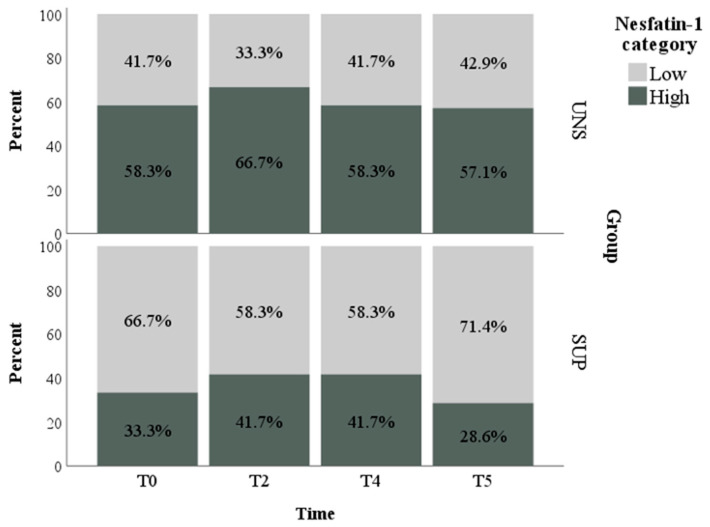
Relative frequencies of Nesfatin-1 categorized into “low” and “high levels” according to group (UNS = control ewes, SUP = supplemented ewes) and time (T0 = before the administration of supplementation; T2 = 20th day of supplementation; T4 = 40th day of supplementation; T5 = 50th day of supplementation).

**Figure 3 animals-11-00682-f003:**
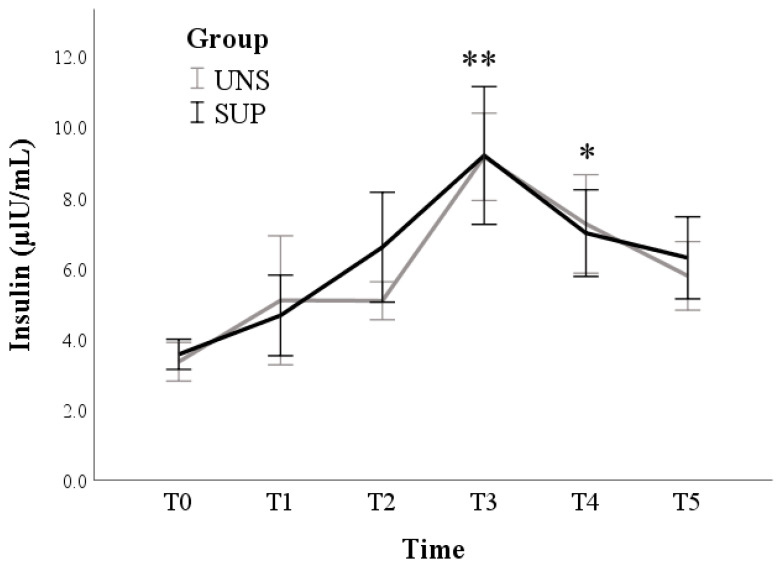
Means and standard errors of insulin plasma concentrations in control (UNS) and supplemented (SUP) lactating sheep monitored before the administration of supplementation (T0), and then every 10 days until dry off (T1–T5). * *p* < 0.05, ** *p* < 0.01: each time vs. T0 in SUP and UNS.

**Figure 4 animals-11-00682-f004:**
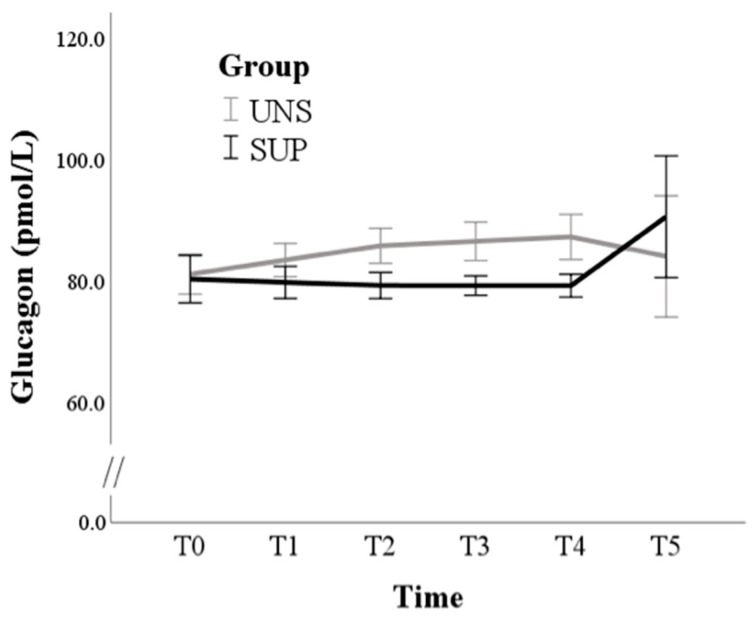
Means and standard errors of glucagon plasma concentrations in control (UNS) and supplemented (SUP) lactating sheep monitored before the administration of supplementation (T0), and then every 10 days until dry off (T1–T5).

**Figure 5 animals-11-00682-f005:**
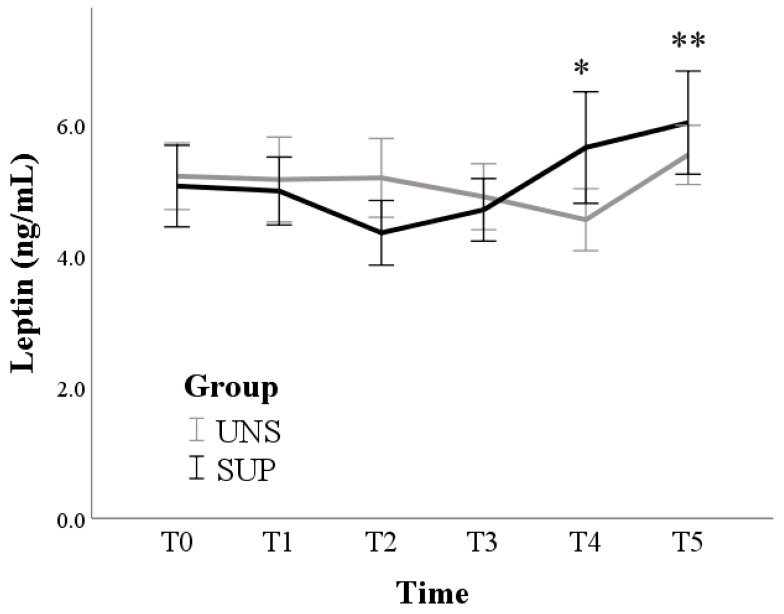
Means and standard errors of leptin plasma concentrations in control (UNS) and supplemented (SUP) lactating sheep monitored before the administration of supplementation (T0), and then every 10 days until dry off (T1–T5). * *p* < 0.05, ** *p* < 0.01: each time vs. T0 in SUP.

**Figure 6 animals-11-00682-f006:**
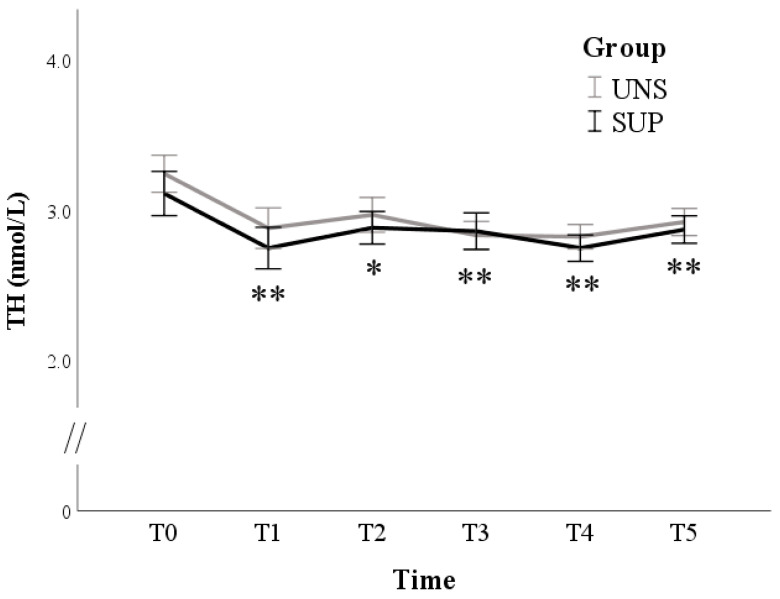
Means and standard errors of 3-3′-5-triiodothyronine (TH) plasma concentrations in control (UNS) and supplemented (SUP) lactating sheep monitored before the administration of supplementation (T0), and then every 10 days until dry off (T1–T5). * *p* < 0.05, ** *p* < 0.01: each time vs. T0 in SUP and UNS.

**Figure 7 animals-11-00682-f007:**
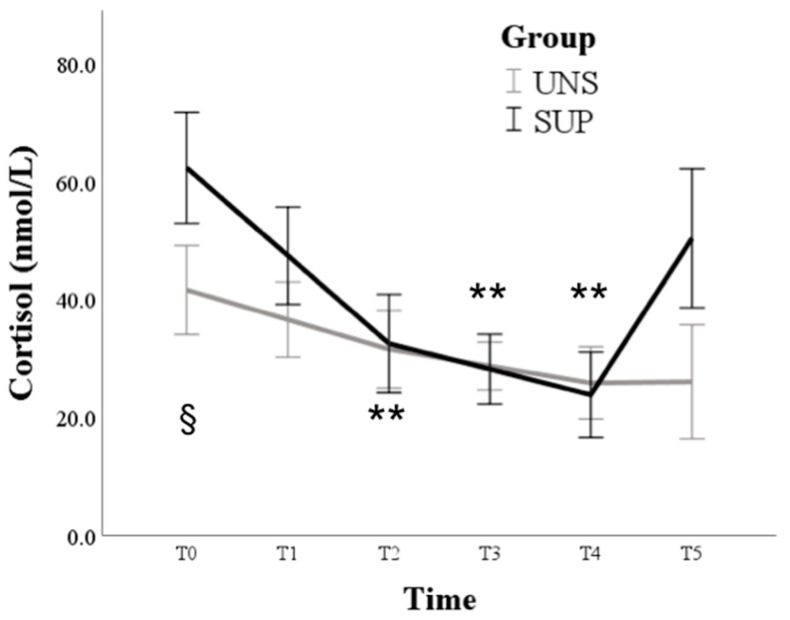
Means and standard errors of cortisol plasma concentrations in control (UNS) and supplemented (SUP) lactating sheep monitored before the administration of supplementation (T0) and then every 10 days until dry off (T1–T5). § *p* < 0.05 UNS vs. SUP; ** *p* < 0.01 vs. T0 in SUP.

**Table 1 animals-11-00682-t001:** Parameters indicating the relative importance of variables in classifying the ewes receiving nutritional supplementation: discriminant loadings, Wilks’ Lambda, and significance of the F test (*p*-value). The variables with a significant F-test are in bold.

Variable	Discriminant Loadings	Wilks’ Lambda	*p* Value
**BCS**	−0.566	0.880	**0.018**
**Nesfatin-1 ***	0.515	0.899	**0.031**
Cortisol	−0.249	0.974	0.288
TH	0.126	0.993	0.588
Insulin #	0.078	0.997	0.737
Leptin	0.042	0.999	0.857
Glucagon	0.037	0.999	0.875

* included as continuous variable; # included after log-transformation.

## Data Availability

Datasets used in the analyses are stored at the authors’ home institution and will be provided upon request.

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
