# Peer review of "Effects of Feed Supplementation on Nesfatin-1, Insulin, Glucagon, Leptin, T3, Cortisol, and BCS in Milking Ewes Grazing on Semi-Natural Pastures"

_animals, 2021, doi:10.3390/ani11030682_

Round 1

Reviewer 1 Report

The document has a substantial improvement but there are still drafting details to correct. 

L18: an article is missing before “semi-extensive” change for “a semi-extensive

L40: Add “the” before “beginning” and before “sexual”

L47: Remove “the” before “drought”

L54: “It was been” is incorrect, change for “it has been”

L56: Add “a” after “as”

L57: “change” should be “changed”

L230: Add “the” before “present”, also remove “a” before “very”

L262: Add “the” before “advancing”, and change “stage” for “stages”

L263: Add a comma after “trail”

L271: There seems to be an extra space between “ewes” and “likely” please verify

L273: Change “concentration” for “concentrations”

L275: Change “concentration” for “concentrations” , and change “was” for “were”, also add “at the” before “40th” and remove the “s” in “days”, leave only one “of lactation” in the sentence

L302: Add “the” before “sexual”

L308: Add a space between “flock” and “which”

L325: Add a space between “on” and “some

Author Response

Dear Editors,

we would like to thank the reviewers for their constructive comments and suggestions, that helped to improve the quality of the manuscript.

Please, find below in red our responses to Reviewers’ comments.

R1

The document has a substantial improvement but there are still drafting details to correct. 

L18: an article is missing before “semi-extensive” change for “a semi-extensive

L40: Add “the” before “beginning” and before “sexual”

L47: Remove “the” before “drought”

L54: “It was been” is incorrect, change for “it has been”

L56: Add “a” after “as”

L57: “change” should be “changed”

L230: Add “the” before “present”, also remove “a” before “very” (now line 274)

L262: Add “the” before “advancing”, and change “stage” for “stages” (now line 306)

L263: Add a comma after “trail” (now line 307)

L271: There seems to be an extra space between “ewes” and “likely” please verify (now line 315)

L273: Change “concentration” for “concentrations” (now line 317)

L275: Change “concentration” for “concentrations” , and change “was” for “were”, also add “at the” before “40th” and remove the “s” in “days”, leave only one “of lactation” in the sentence (now line 319)

L302: Add “the” before “sexual” (now line 346)

L308: Add a space between “flock” and “which” (now line 352)

L325: Add a space between “on” and “some (now line 369)

All the suggestions were accepted

Reviewer 2 Report

Thanks for the changes made, for me the paper has been clear.

I have three suggestions to make:

-Line 51: Change worming to warming

-Lines 121-124: Dairy production data, as well as Figure S2, should be in Results and not in Materials and Methods as they are now

-Line 197: You have eliminated the entire footnote of Figure 1. You must restore it.

Author Response

Dear Editors,

we would like to thank the reviewers for their constructive comments and suggestions, that helped to improve the quality of the manuscript.

Please, find below in red our responses to Reviewers’ comments.

R2

Thanks for the changes made, for me the paper has been clear.

I have three suggestions to make:

-Line 51: Change worming to warming

Done

-Lines 121-124: Dairy production data, as well as Figure S2, should be in Results and not in Materials and Methods as they are now

We added in the results the following sentence: “As previously reported (28), milk production at the three sampling times was: 297±63 mL by UNS and 306±64 mL by SUP ewes, 134±47 mL by UNS and 240±52 mL by SUP ewes and 125±51 mL by UNS and 192±52 mL by SUP ewes, respectively (Figure S2).” 

We do not agree with the reviewer and we believe it is more correct to leave FigS2 in the supplementary file as the data refer to a previously published work (28)

-Line 197: You have eliminated the entire footnote of Figure 1. You must restore it.

We have restored it

This manuscript is a resubmission of an earlier submission. The following is a list of the peer review reports and author responses from that submission.

Round 1

Reviewer 1 Report

Regarding manuscript # animals-1077040 entitled “Effects of food supplementation on some metabolic hormones and BCS in milking ewes grazing on semi-natural pastures”, the authors aimed to evaluate the effects of cereals supplementation on body condition score and metabolic hormones profile in milking ewes grazing on semi- natural pastures on mid mountain rangelands.

The novelty of the study is weak. Additionally, the design of the experiment is not correct. The study was done on milking ewes. The authors must focus in ewes’ milk production during the experimental period. The amount of milk yield can greatly affect the BCS. Additionally, nothing was available in data about feed intake which can also affect BCS and metabolic hormones.  

Did the authors think that “600 g/day/animal of corn and barley” is considered a feed supplementation”?!

Did the authors feed these animals individually? Did all animals in SUP group consumed 600 g/day/animal of corn and barley?

What was the parity and age of these ewes?

The manuscript needs extensive English editing.

“Food” is used for human while “feed” is used for animals.

The section “2.2. Blood collection and assays” each paragraph has one or two sentence. Please merge these paragraphs in 2-3 paragraphs.

The title is general and not clears what the authors studied in their experiment.

Reviewer 2 Report

There are some issues that need to be addressed before it can be published such as:

It is advisable to review the INTRODUCTION in order to make it more specific with the research, I don’t consider that explaining each hormone (or when they were identified) is necessary, it would be better to just adhere to why they might be important and justify the research accordingly.

There are also some minor grammar mistakes that need correction, such as:

L49: The verb “are” does not seem to agree with the subject, it should be changed to “is” or “change” could be “changes”

L68: “carbohydrate” and “protein” should be plural as they don’t agree in number with the other words of the phrase

L75: The preposition “to” is missing after “due”

L83: There is a preposition missing between “role” and “stimulating”, could be “in, when” or one of your choosing that better explains what you mean

L108-110: This sentence seems a little incomplete, some words are needed to make it clearer, please rewrite.

L122: The word “was” is missing before “fed”

L123: Change “were” for “was”, the verb should be singular

L131: there is a missing word between “then” and “every”, could be “evaluated”.

L236: The verbs don’t agree in number, remove “a” before “very” to make the phrase plural so it agrees with “two ewe groups”

L242: Rewrite the sentence “Considering as relatively equal the milk yield on…” you could either add some commas or rewrite something like: Milk yield was considered as relatively equal, on day T0, it….

L279: The word “two” is missing the “t” it reads “wo”

L312: There seems to be a missing comma after “stress”

L320: After “August” it says “et at”, could be “and at”

L322: Add “it” between “so” and “was”

L324-325: the sentence is not clear, please rewrite.

L333: Change “they” for “their”

L335: Change “difficult” for “difficulties”

Reviewer 3 Report

My comments are in the attached file.
